# OpenReview forum: "Extragradient Preference Optimization (EGPO): Beyond Last-Iterate Convergence for Nash Learning from Human Feedback"
_colmweb.org/COLM/2025/Conference — COLM 2025_

### Official Review · Reviewer_PkxM · 2025-05-10

**Rating:** 6
**Confidence:** 3
**Ethics Flag:** 1

**Summary:**

The paper introduces EGPO, an algorithm for Nash Learning from Human Feedback (NLHF), targeting non-transitive human preferences. Traditional RLHF assumes transitive preferences via the Bradley-Terry model, which fails for aggregated human preferences. EGPO claims to achieve last-iterate linear convergence to the Nash equilibrium (NE) of KL-regularized games and polynomial convergence for unregularized games. It avoids nested optimization via an online IPO formulation and demonstrates empirical superiority in language model alignment tasks.

**Questions To Authors:**

Could you please provide more experiments on diverse datasets and models? In particular for recent open-source models. Also I think it's better to show the analysis of the time cost during training.

**Reasons To Accept:**

Theoretical Rigor: Provides convergence guarantees (last-iterate linear/polynomial rates) for both regularized and original games, addressing a critical gap in prior NLHF methods.

Algorithmic Innovation: Derives an online IPO loss to eliminate nested optimization, enabling faithful neural network implementation.

Empirical Validation: Shows improved win-rates over baselines (e.g., Nash-MD, MPO) on synthetic and language model tasks.

**Reasons To Reject:**

Limited Scalability:Experiments use small models (gemma-2-2b-it) and a single dataset (PKU-SafeRLHF). No evidence is provided for scalability to state-of-the-art open source LLMs (e.g., LLaMA series, Qwen series) or diverse preference datasets.

Incomplete Baseline Comparison: In the experiment the Nash-MD-PG’s poor performance is a little weird, I think here we need more details to explain why the method is poor in this setting, and why it only converges when $\beta$ is large?

Practical Relevance:  No discussion of computational costs or sample efficiency compared to other RLHF methods. In particular the method is designed for non-transitive case, but I am also interesting for the transitive case the comparison of EGPO with PPO or GRPO.  Also is it possible to provide the code for the LLM training?

Incremental Contribution: EGPO builds heavily on Cen et al. (2021)’s extragradient methods and Munos et al. (2023)’s NLHF framework. The online IPO formulation, while useful, is a technical extension rather than a conceptual breakthrough.

---

> ### Author Response · Authors · 2025-06-02
> **Reply to Reviewer PkxM**
>
> ### Reasons To Reject:
>
> **[R1]** Limited Scalability:Experiments use small models (gemma-2-2b-it) and a single dataset (PKU-SafeRLHF). No evidence is provided for scalability to state-of-the-art open source LLMs (e.g., LLaMA series, Qwen series) or diverse preference datasets.
>
> [Reply] Thank you for pointing this out. Due to resource constraints, we were limited to using a 2B model with LoRA. However, our algorithm design and implementation are model/data-agnostic (only requiring generating responses, calculating log probabilities, and gradient backup and restoration), and our theoretical guarantees outperform those of all baselines. We believe that given the same base model and dataset, EGPO will outperform the baselines mentioned in our work.
>
> **[R2]** Incomplete Baseline Comparison: In the experiment the Nash-MD-PG’s poor performance is a little weird, I think here we need more details to explain why the method is poor in this setting, and why it only converges when $\beta$ is large?
>
> [Reply] We clarify this here. For benchmarks, we use the standard setting of $\beta=0.1$ and $\eta=5\times 10^{-7}$ from previous RLHF works. The Nash-MD-PG trainer is directly taken from the TRL library (`trl.NashMDTrainer`), so its slow convergence and poor behavior simply verify its theoretical guarantee of a constant gap of $\tilde{O} (\eta/\beta)$ (see Table 1). When running numerical simulations, the $\eta$ values reported in the titles of Figures 2 to 4 are those passed into the optimizer. If we denote $\eta_\textup{theory}$ as the $\eta$ in Theorem 1, they are actually $\frac{\eta_\textup{theory} \beta |\mathcal{Y}|}{4}$ according to Equations (4) and (5). Therefore, $\eta_\textup{theory}/\beta \propto \eta / \beta^2$. Hence, for $\beta = 0.001$ and $0.01$, Nash-MD-PG does not converge as well as the other algorithms.
>
> **[R3]** Practical Relevance: No discussion of computational costs or sample efficiency compared to other RLHF methods. In particular the method is designed for non-transitive case, but I am also interesting for the transitive case the comparison of EGPO with PPO or GRPO. Also is it possible to provide the code for the LLM training?
>
> [Reply] For computational costs, please refer to our reply to all reviewers.
>
> We also clarify that beyond the capability to handle non-transitive preferences, NLHF differs from standard RLHF as a **game-theoretic** formulation with distinct objectives. NLHF algorithms optimize the **worst-case win-rate against any policy**, while standard RLHF algorithms optimize **expected rewards**. We will discuss the motivational differences between NLHF and RLHF more clearly in the paper.
>
> When comparing algorithms by win-rates, EGPO policies achieve **at least** $50\\%$ (typically higher) win probability against PPO/DPO policies. However, when comparing by rewards, PPO/DPO policies are more likely to outperform EGPO policies.
>
> **[R4]** Incremental Contribution: EGPO builds heavily on Cen et al. (2021)’s extragradient methods and Munos et al. (2023)’s NLHF framework. The online IPO formulation, while useful, is a technical extension rather than a conceptual breakthrough.
>
> [Reply] Generalizing theoretical algorithms to LLM applications is non-trivial. While [1] initiated the study of NLHF, the theoretical algorithm Nash-MD is essentially a variant of mirror descent. MPO [2] is also an LLM implementation of a previous algorithm, MMD [3]. The value of NLHF studies lies in identifying the theory-practice gap and proposing better approximations and sampling schemes to reduce bias when theoretical algorithms are applied to LLMs. In this regard, the online IPO formulation significantly improves upon previous nested optimization paradigms by reducing both algorithm complexity and optimization bias. We therefore believe this formulation is significant to NLHF research, as it can be easily applied to previous algorithms.
>
>
> ### Questions To Authors:
>
> **[Q1]** Could you please provide more experiments on diverse datasets and models? In particular for recent open-source models. Also I think it's better to show the analysis of the time cost during training.
>
> [Reply] Due to resource constraints, we may not provide results on other datasets and models before the stage 1 rebuttal deadline. We are currently training `Qwen/Qwen2.5-1.5B-Instruct` using a subset of `OpenRLHF/prompt-collection-v0.1` (the same dataset as Appendix C.3 of [2]). As suggested by reviewer DDAh, we also aim to complete experiments under transitive preference settings before the camera-ready deadline if our work is accepted. Time cost analysis is provided in the reply to all reviewers.
>
> [1] Nash learning from human feedback
>
> [2] Magnetic preference optimization: Achieving last-iterate convergence for language model alignment
>
> [3] A unified approach to reinforcement learning, quantal response equilibria, and two-player zero-sum games

---

> > ### Comment · Reviewer_PkxM · 2025-06-08
> >
> > Thanks for your update on the theory of the method and I have changed my score

---

> > > ### Author Response · Authors · 2025-06-08
> > >
> > > Thank you for your support!

---

### Official Review · Reviewer_GcQP · 2025-05-11

**Rating:** 8
**Confidence:** 4
**Ethics Flag:** 1

**Summary:**

This paper proposes EGPO, a new algorithm for aligning language models with non-transitive human preferences in RLHF. Unlike traditional methods, EGPO frames the task as a game and achieves last-iterate linear convergence to the Nash equilibrium without nested optimization. It outperforms existing baselines in empirical tests and offers a practical, gradient-based solution for neural networks. The paper also suggests future directions with theoretical justification to improve convergence speed and reduce sample complexity.

**Questions To Authors:**

1. Line 308 and beyond elaborated on some limitations of this work. They are future works that make things more efficient, but not 'limitations'. The limitations would be in the problem setting and data resources in Nash Learning from Human Feedback. Taking a step back, does this work have a better coverage or push the boundary of the original proposal of NLHF in terms of problem setting and data resources?

**Reasons To Accept:**

- This paper presents theoretical soundness for general preference alignment: EGPO achieves last-iterate linear convergence to the Nash equilibrium in KL-regularized games and is robust to gradient noise, with convergence rates unaffected.
- Sufficient experimental evidence with practical Implementation is presented in a comprehensive manner: EGPO avoids nested optimization via an equivalent gradient-based formulation and outperforms baselines in benchmark win-rate evaluations.

**Reasons To Reject:**

- In the online RLHF setting, this paper presented many experimental results in the appendix, while the main body lacks a good reasoning and summary of the performance impact. There are some remarks in Line 292 and beyond, however, it could be more interesting if the revealed good cases are explained with a why, and the limitation of the proposed method are also explicitly presented to readers.

---

> ### Author Response · Authors · 2025-06-02
> **Reply to Reviewer GcQP**
>
> ### Reasons To Reject:
>
> **[R1]** In the online RLHF setting, this paper presented many experimental results in the appendix, while the main body lacks a good reasoning and summary of the performance impact. There are some remarks in Line 292 and beyond, however, it could be more interesting if the revealed good cases are explained with a why, and the limitation of the proposed method are also explicitly presented to readers.
>
> [Reply] Thanks for the suggestion! We plan to revise Section 5.2 as follows: shorten the experiment setup descriptions by mentioning only the model and dataset, with all other details moved to Appendix E.3.1; add more discussion on approximating uniform sampling, including hyperparameter search for (temperature, top_k); in the remarks, discuss EGPO's efficiency (though it requires two iterations per update step, it can achieve even better results using half the number of iterations compared to baselines) and relate this to theoretical findings and numerical simulations.
>
> ### Questions To Authors:
> **[Q1]** Line 308 and beyond elaborated on some limitations of this work. They are future works that make things more efficient, but not 'limitations'. The limitations would be in the problem setting and data resources in Nash Learning from Human Feedback. Taking a step back, does this work have a better coverage or push the boundary of the original proposal of NLHF in terms of problem setting and data resources?
>
> [Reply] Regarding the problem setting, directly learning a preference model may result in preference hacking, as any misspecifications or biases in the preference model may impact the policy model. In our work, we avoid this problem by assuming the preference model is the ground truth, but in practice, we need to query humans directly for the ground truth during evaluation. This work aims to deliver an algorithm with better theoretical guarantees and implementation accuracy, so we do not change the boundary of NLHF problems.

---

> > ### Comment · Reviewer_GcQP · 2025-06-08
> >
> > Thank you for sharing your planned updates and candid response to my questions. I maintain my score.

---

> > > ### Author Response · Authors · 2025-06-09
> > >
> > > Thank you for your support!

---

### Official Review · Reviewer_Z3uq · 2025-05-12

**Rating:** 7
**Confidence:** 3
**Ethics Flag:** 1

**Summary:**

This work introduces EGPO, which reformulates extragradient preference optimization as a single online IPO gradient, giving noise-robust last-iterate linear convergence and faster empirical alignment of Gemma-2B and bandits versus baselines.

**Questions To Authors:**

Could the authors provide the sensitive analysis of $\beta / \eta$?

**Reasons To Accept:**

1. Guarantees last-iterate linear convergence to the KL-regularised Nash equilibrium, holding even under sub-Gaussian gradient noise.
2. Re-formulates extragradient as a single online IPO gradient, removing costly inner loops and aligning theory and code.
3.  Matches/outperforms baselines on tabular bandits and boosts win-rate in Gemma-2-2B alignment under identical training budgets.

**Reasons To Reject:**

1. EGPO needs two forward/backward passes per update and relies on approximate uniform sampling of candidate responses, which can double the compute cost and introduce bias.
2. The paper offers little analysis of $\beta / \eta$ sensitivity or wall-clock efficiency relative to single-step methods

---

> ### Author Response · Authors · 2025-06-02
> **Reply to Reviewer Z3uq**
>
> ### Reasons To Reject:
>
> **[R1]** EGPO needs two forward/backward passes per update and relies on approximate uniform sampling of candidate responses, which can double the compute cost and introduce bias.
>
> [Reply] These points are important. With two forward/backward passes per update, Tables 2 and 7 show that EGPO running for 5 epochs outperforms all other algorithms running for 10 epochs, suggesting that EGPO's iteration complexity can be reduced to half that of other algorithms. Regarding the approximation of uniform sampling, we acknowledge that it introduces bias. In practice, we searched over configurations of (temperature, top_k) and selected the optimal configuration of $(2, 10)$ for all algorithms.
>
> **[R2]** The paper offers little analysis of $\beta/\eta$ sensitivity or wall-clock efficiency relative to single-step methods.
>
> [Reply] We experimented with different values of $\beta$ and $\eta$ for the numerical simulations and found that when $\beta$ is small, the optimal $\eta$ is also small. In our numerical simulations, the $\eta$ values reported in the titles of Figures 2 to 4 are those passed into the optimizer. If we denote $\eta_\textup{theory}$ as the $\eta$ in Theorem 1, these values are actually $\frac{\eta_\textup{theory} \beta |\mathcal{Y}|}{4}$ according to Equations (4) and (5). Therefore, the sensitivity of $\eta$ to $\beta$ is expected. Due to computational resource constraints, we have not experimented with different $\beta$ and $\eta$ values for LLM benchmarks.
>
> Regarding wall-clock efficiency, please refer to our reply to all reviewers.
>
> ### Questions To Authors:
>
> **[Q1]** Could the authors provide the sensitive analysis of $\beta/\eta$?
>
> [Reply] Please see the response to **[R2]**.

---

> > ### Comment · Reviewer_Z3uq · 2025-06-08
> > **Official Response by Reviewer Z3uq**
> >
> > Thank you for the response; all my concerns have been addressed. I would like to keep my rating.

---

> > > ### Author Response · Authors · 2025-06-08
> > >
> > > Thank you for your support!

---

### Official Review · Reviewer_DDAh · 2025-05-12

**Rating:** 7
**Confidence:** 4
**Ethics Flag:** 1

**Summary:**

This paper introduces Extragradient Preference Optimization (EGPO), a novel algorithm for Nash Learning from Human Feedback (NLHF). Traditional Reinforcement Learning from Human Feedback (RLHF) methods often assume transitive human preferences, which may not hold true at a populational level. NLHF addresses this by aiming to find the Nash Equilibrium (NE) of the game defined by human preferences.

EGPO is presented as an algorithm that achieves last-iterate linear convergence to the NE of KL-regularized games and polynomial convergence to the NE of original games. It is also designed to be robust to noise. A key contribution is an equivalent practical implementation of EGPO using gradients of an online variant of the Identity Preference Optimization (IPO) loss. This approach avoids the nested optimization common in previous methods, allowing for a more faithful implementation with neural networks. The paper's empirical evaluations demonstrate EGPO's improvements compared to baseline methods in terms of pairwise win-rates, using ground truth preferences, when trained for the same number of epochs. These results suggest EGPO's theoretical strengths and practical advantages for aligning language models with non-transitive human preferences.

The paper is of high quality, presenting a novel algorithm with strong theoretical backing and empirical validation. The methodology is well-described, and the problem it addresses is significant in the field of RLHF and LLM alignment. The paper is generally well-written and structured. The introduction clearly outlines the limitations of existing methods and the motivation for the proposed approach. The contributions are explicitly stated. The online IPO formulation enhances practical clarity.

The proposed EGPO algorithm and its equivalent online IPO formulation appear to be original contributions. The paper builds upon existing work in game-theoretic RLHF  and methods for computing equilibria in zero-sum games  but offers a novel combination and a practical implementation strategy that avoids common pitfalls like nested optimization. This work holds considerable significance for the LLM alignment community. Addressing non-transitive human preferences is a crucial step towards more robust and realistic preference modeling. The achievement of last-iterate convergence is practically important for training large models.

**Questions To Authors:**

- Could you elaborate on the computational overhead of EGPO compared to the baselines, particularly in the LLM alignment experiments (e.g., wall-clock time per epoch, memory usage)?
The online IPO formulation relies on specific choices for $ρ$ and $μ$ (e.g., $π^s =Uniform(Y)×Uniform(Y)$ in Equation 4 and 5). Have you explored other choices for these distributions, and how might they affect performance or the equivalence to the theoretical EGPO updates?

- Regarding the approximation for uniform sampling from the response space: How sensitive are the results to the specific top-k and temperature values used? Could this approximation introduce any systematic biases that might affect certain types of prompts or responses differently?

- In Table 1, EGPO is shown to converge to the regularized QRE at a rate of $$ \tilde{O}((1−ηβ)^T) $$
How does the choice of the regularization coefficient β practically impact the convergence speed and the quality of the final policy in the LLM experiments? The paper mentions $β=0.001$ is the most challenging case in simulations  and $β=0.1$ is used in NLHF training.

- The paper notes that MPO, while having lower win-rates against the reference policy than Nash-MD, consistently beat Nash-MD in direct pairwise comparisons, indicating non-transitivity in the ground truth preference P. Does EGPO show similar robustness or advantages in highly non-transitive scenarios compared to methods that might implicitly favor transitivity?

**Reasons To Accept:**

- EGPO offers last-iterate linear convergence for KL-regularized games and polynomial convergence for original games, which are significant theoretical contributions in NLHF. The robustness to noise is also a valuable theoretical property.
- The derivation of an equivalent online IPO formulation is a key strength. This avoids nested optimization loops, which are problematic in practice and can lead to discrepancies between theory and implementation. This makes EGPO more readily applicable to large neural network models.
- The problem of non-transitive populational human preferences in RLHF is a known limitation of traditional Bradley-Terry models, and is timely to study. Finding the Nash Equilibrium of these preference games is a more principled approach than prior works.
- The experimental results demonstrate that EGPO outperforms several baseline methods in terms of pairwise win-rates against a ground truth preference model when trained for the same number of epochs.
- The focus on last-iterate convergence is highly relevant for LLMs, as averaging historical policies is often infeasible due to computational and storage constraints.

**Reasons To Reject:**

- While the online IPO formulation simplifies things, the overall practical implementation complexity across different LLM architectures and datasets might still be a concern. The paper uses gemma-2-2b-it; performance on significantly larger or different types of models would be interesting.
- The paper acknowledges that directly sampling uniformly from the response space is impractical for auto-regressive models and uses an approximation by generating responses with top-k and temperature settings. The impact of this approximation on the theoretical guarantees and empirical performance could be further explored or discussed.
- The performance of EGPO, like many RL algorithms, might be sensitive to hyperparameter choices (e.g., $β$, $η$). While the paper presents results for specific settings, a more extensive ablation study on hyperparameter sensitivity could strengthen the practical claims.
- The experiments use a fine-tuned language model as the "ground truth preference". While a common practice due to resource constraints, the characteristics of this surrogate (e.g., its own potential biases or non-transitivity) and how they might interact with EGPO's evaluation could be discussed. The paper does note this model can be replaced with human annotators or other LLMs.
- The paper motivates NLHF by the non-transitive nature of populational human preferences. It would be beneficial to discuss the extent or types of non-transitivity EGPO is particularly well-suited to handle, perhaps with more varied non-transitive scenarios in the numerical simulations. The paper mentions the ground truth preference $P$ demonstrates non-transitive behavior in one instance.

---

> ### Author Response · Authors · 2025-06-02
> **Reply to Reviewer DDAh (2/2)**
>
> ### Questions To Authors:
>
> **[Q1]** Could you elaborate on the computational overhead of EGPO compared to the baselines, particularly in the LLM alignment experiments (e.g., wall-clock time per epoch, memory usage)? The online IPO formulation relies on specific choices for $\rho$ and $\mu$ (e.g., Unif * Unif in Equation 4 and 5). Have you explored other choices for these distributions, and how might they affect performance or the equivalence to the theoretical EGPO updates?
>
> [Reply] Regarding computational overhead, please refer to the reply to all reviewers.
>
> We have not explored other configurations of $\rho$ and $\mu$ in the theoretical analysis, as the **current choice is sufficient for deriving the equivalence**. We hypothesize that $\rho$ cannot be changed, as $\Sigma(\rho)$ (see line 669) would be not identical to $I + C \cdot \mathbb{1}$ otherwise. In practice, as mentioned in previous replies, we tested several configurations of (temperature, top_k) to approximate $\rho$, and the combination of $(2, 10)$ performed best.
>
> **[Q2]** Regarding the approximation for uniform sampling from the response space: How sensitive are the results to the specific top-k and temperature values used? Could this approximation introduce any systematic biases that might affect certain types of prompts or responses differently?
>
> [Reply] We searched over a small space of (temperature, top_k), and the win-rates of Online IPO 1 (OMD) against $\pi_\textup{ref}$ varied between $65\%$ and $72\%$, with $72\%$ corresponding to $(2, 10)$. While the choice of parameters shows some sensitivity, it is not particularly high. We observed no extremely abnormal responses from these configurations. However, when temperature $=10$ and top_k $=20$, responses contained highly irrelevant sentences.
>
> **[Q3]** In Table 1, EGPO is shown to converge to the regularized QRE at a rate of $\tilde{O} ((1-\eta\beta)^T)$. How does the choice of the regularization coefficient β practically impact the convergence speed and the quality of the final policy in the LLM experiments? The paper mentions $\beta=0.001$ is the most challenging case in simulations and $\beta=0.1$ is used in NLHF training.
>
> [Reply] As $\beta$ increases, the convergence rate accelerates since $1 - \eta \beta$ decreases. However, this also means the final policy becomes more similar to the reference policy, achieving less human preference alignment. From the duality gap perspective, $(\textup{DualGap} - \textup{DualGap}_\beta) \propto \beta$; larger $\beta$ values make the QRE deviate further from the actual NE. In the extreme case where $\beta \to 0$, the regularized game closely approximates the original game (which may have multiple NEs), making it harder to solve. The choice of $\beta=0.1$ or larger is common in the RLHF/NLHF field.
>
> **[Q4]** The paper notes that MPO, while having lower win-rates against the reference policy than Nash-MD, consistently beat Nash-MD in direct pairwise comparisons, indicating non-transitivity in the ground truth preference P. Does EGPO show similar robustness or advantages in highly non-transitive scenarios compared to methods that might implicitly favor transitivity?
>
> [Reply] We have not observed this phenomenon in our Safe-RLHF experiments, as EGPO consistently achieves the highest win-rates against both $\pi_\textup{ref}$ and other policies. To verify this finding, we could replace the preference model with a reward model (following standard RLHF) and rerun the experiments. However, due to resource constraints, we will complete this analysis before the camera-ready deadline if our work is accepted.

---

> > ### Comment · Reviewer_DDAh · 2025-06-09
> >
> > Thank you very much for the detailed replies. I am more certain in the paper now and have slightly increased my rating.

---

> > > ### Author Response · Authors · 2025-06-09
> > >
> > > Thank you for your support!

---

> ### Author Response · Authors · 2025-06-02
> **Reply to Reviewer DDAh (1/2)**
>
> ### Reasons To Reject:
>
> **[R1]** While the online IPO formulation simplifies things, the overall practical implementation complexity across different LLM architectures and datasets might still be a concern. The paper uses gemma-2-2b-it; performance on significantly larger or different types of models would be interesting.
>
> [Reply] The implementation is model-agnostic. Our code inherits the `OnlineDPOTrainer` class from the TRL library and only requires (1) generating responses, (2) calculating log probabilities, and (3) gradient backup and restoration. Since all these operations are high-level, no architecture- or dataset-specific handling is needed.
>
> **[R2]** The paper acknowledges that directly sampling uniformly from the response space is impractical for auto-regressive models and uses an approximation by generating responses with top-k and temperature settings. The impact of this approximation on the theoretical guarantees and empirical performance could be further explored or discussed.
>
> [Reply] This is worth investigating. A similar situation of sampling from a uniform distribution was encountered in previous work [2]. In that paper, the authors treat the uniform distribution as a posterior on the current policy $\pi_\theta$ and sample directly from $\pi_\theta$. However, neither our work nor theirs can provide theoretical justification for sampling from $\pi_\theta$.
>
> We tested all combinations of temperature $\in \\{1,2,5\\}$ and top_k $\in \\{5, 10\\}$. The combination of $(2, 10)$ performed best in terms of win-rates against $\pi_\textup{ref}$.
>
> **[R3]** The performance of EGPO, like many RL algorithms, might be sensitive to hyperparameter choices (e.g., $\beta$, $\eta$). While the paper presents results for specific settings, a more extensive ablation study on hyperparameter sensitivity could strengthen the practical claims.
>
> [Reply] We agree that thorough hyperparameter investigations are important. For fair comparison with baselines, we adopt standard benchmark values of $\beta=0.1$ and $\eta=5\times 10^{-7}$. In numerical simulations (Appendix E.2.2, Figures 2 to 4), we use different $\beta$ settings, with $\eta$ values determined through hyperparameter search for each $\beta$.
>
> **[R4]** The experiments use a fine-tuned language model as the "ground truth preference". While a common practice due to resource constraints, the characteristics of this surrogate (e.g., its own potential biases or non-transitivity) and how they might interact with EGPO's evaluation could be discussed. The paper does note this model can be replaced with human annotators or other LLMs.
>
> [Reply] We clarify this point here. As stated in Section 4.1, our study does not involve "preference modeling" since our focus is on the optimization side -- specifically, how quickly we can achieve a specific NE. Consequently, we simply assume the preference model is accurate. In Section 8.1 of [2], the authors demonstrate that increasing preference model capacity improves accuracy, though the gain from 3B to 11B is minimal. We expect similar findings: if we incorporate preference modeling using a larger model to learn preferences, bias will decrease, but gains will be diminishing.
>
> **[R5]** The paper motivates NLHF by the non-transitive nature of populational human preferences. It would be beneficial to discuss the extent or types of non-transitivity EGPO is particularly well-suited to handle, perhaps with more varied non-transitive scenarios in the numerical simulations. The paper mentions the ground truth preference $P$ demonstrates non-transitive behavior in one instance
>
> [Reply] This question is central to the motivation of NLHF works. EGPO assumes no special structure on $\mathcal{P}$, enabling it to handle any extent or type of non-transitivity with theoretical guarantees. For detailed motivation on non-transitive preferences, see Section 3.2 of [2].
>
> [1] The Crucial Role of Samplers in Online Direct Preference Optimization
>
> [2] Nash Learning from Human Feedback

---

### Author Response · Authors · 2025-06-02
**Reply to All Reviewers: Discussion on Computation Overhead**

When running on $8\times$A6000 GPUs, one epoch takes Online IPO 1 (OMD) $1.51$ hrs, Online IPO 2 $0.98$ hrs, NashMD $3.48$ hrs, NashMD-PG $4.48$ hrs, and EGPO $1.56$ hrs (one effective epoch takes $2\times$ time). Online IPO 2 is the most time-efficient, as it requires only $2$ (v.s. $3$) rollouts per prompt due to a variance reduction technique. NashMD and NashMD-PG are less efficient because they require sampling from a geometric mixture of two policies.

When using the same micro batch size of $8$, EGPO consumes around $33$G of GPU memory per GPU, while all other algorithms consume around $28$G. This difference occurs because EGPO backs-up gradients and optimizer states.

---

### Author Response · Authors · 2025-06-02
**Reply to All Reviewers: Additional Model Benchmark**

We trained `Qwen/Qwen2.5-1.5B-Instruct` using a subset of `OpenRLHF/prompt-collection-v0.1` (the same dataset used in Appendix C.3 of [1]), and evaluated using a disjoint subset of the same dataset. Since the code in [1] does not provided support for this dataset (I think they missed this part when submitting the code) and their support for Safe-RLHF is hardcoded, we skipped MPO in this additional experiment. Below are our results. It can be seen that only Online IPO 2, Nash-MD (not -PG), and EGPO are able to get non-trivial win-rates. EGPO still outperforms all the baselines.

| ALG | Ep | $\pi_\text{ref}$ | OIPO1 |  | OIPO2 |  | NMD |  | NMDPG |  | EGPO |  |
| --- | --- | --- | --- | --- | --- | --- | --- | --- | --- | --- | --- | --- |
|  |  |  | 7 | 9 | 5 | 8 | 6 | 7 | 2 | 4 | 8 | 10 |
| OIPO1 | 7 | **50.5%** |  |  | 49.8% | 49.8% | 49.8% | 49.8% | **51.3%** | **51.1%** | 45.5% | 44.8% |
|  | 9 | **50.1%** |  |  | 49.1% | 49.8% | 49.4% | 49.4% | **52.4%** | **52.2%** | 46.8% | 47.0% |
| OIPO2 | 5 | 49.1% | **50.2%** | **50.9%** |  |  | 49.5% | 49.6% | **51.9%** | **51.7%** | 46.0% | 45.5% |
|  | 8 | **50.3%** | **50.2%** | **50.2%** |  |  | 49.2% | 49.5% | **52.5%** | **51.1%** | 45.7% | 45.0% |
| NMD | 6 | **50.7%** | **50.2%** | **50.6%** | **50.5%** | **50.8%** |  |  | **52.5%** | **51.1%** | 44.7% | 46.2% |
|  | 7 | **51.7%** | **50.2%** | **50.6%** | **50.4%** | **50.5%** |  |  | **52.0%** | **51.8%** | 47.0% | 46.8% |
| NMDPG | 2 | 49.1% | 48.7% | 47.6% | 48.1% | 47.5% | 47.5% | 48.0% |  |  | 44.0% | 43.1% |
|  | 4 | 49.5% | 48.9% | 47.8% | 48.3% | 48.9% | 48.9% | 48.2% |  |  | 43.6% | 44.0% |
| EGPO | 8 | **54.2%** | **54.5%** | **53.2%** | **54.0%** | **54.3%** | **55.3%** | **53.0%** | **56.0%** | **56.4%** |  |  |
|  | 10 | **54.2%** | **55.2%** | **53.0%** | **54.5%** | **55.0%** | **53.8%** | **53.2%** | **56.9%** | **56.0%** |  |  |

| ALG | Ep | $\pi_\text{ref}$ | ALG | Ep | $\pi_\text{ref}$ | ALG | Ep | $\pi_\text{ref}$ | ALG | Ep | $\pi_\text{ref}$ | ALG | Ep | $\pi_\text{ref}$ |
| --- | --- | --- | --- | --- | --- | --- | --- | --- | --- | --- | --- | --- | --- | --- |
| OIPO1 | 1 | 50.2% | OIPO2 | 1 | 47.9% | NMD | 1 | 49.7% | NMDPG | 1 | 49.1% | EGPO | 1 | 49.8% |
|  | 2 | 50.8% |  | 2 | 49.7% |  | 2 | 51.6% |  | 2 | **50.1%** |  | 2 | 51.2% |
|  | 3 | 49.8% |  | 3 | 50.4% |  | 3 | 51.6% |  | 3 | 49.5% |  | 3 | 50.0% |
|  | 4 | 48.3% |  | 4 | 49.1% |  | 4 | 51.4% |  | 4 | **51.9%** |  | 4 | 52.4% |
|  | 5 | 50.2% |  | 5 | **51.4%** |  | 5 | 50.7% |  | 5 | 50.1% |  | 5 | 54.4% |
|  | 6 | 50.5% |  | 6 | 50.7% |  | 6 | **51.7%** |  | 6 | 49.6% |  | 6 | 54.7% |
|  | 7 | **51.2%** |  | 7 | 50.3% |  | 7 | **53.2%** |  | 7 | 50.0% |  | 7 | 54.2% |
|  | 8 | 50.1% |  | 8 | **52.5%** |  | 8 | 51.3% |  | 8 | 50.1% |  | 8 | **55.1%** |
|  | 9 | **50.9%** |  | 9 | 50.8% |  | 9 | 51.0% |  | 9 | 49.7% |  | 9 | 54.2% |
|  | 10 | 50.7% |  | 10 | 51.4% |  | 10 | 50.4% |  | 10 | 48.5% |  | 10 | **55.0%** |


[1] Magnetic preference optimization: Achieving last-iterate convergence for language model alignment

---

### Author Response · Authors · 2025-06-05
**Update to Range of $\eta$ in Theorems 1, 2, and 4**

By fixing some loose ends, we now can sharp the range for learning rate: $\eta \le \frac{1}{\beta + 3}$ in Theorems 1, 2, and 4. The original range was $\eta \le \frac{1}{\beta + 2 |\mathcal{Y}| + 1}$ where $|\mathcal{Y}|$ is the number of actions. This eliminates polynomial dependence on action size, which could be very large for NLP tasks.

---

### Author Response · Authors · 2025-06-06

Dear reviewers,

Thank you for your effort in the reviewing process! We updated our main theorem to be more inclusive for learning rates, posted additional benchmark results, and provided discussions on the computation overheads. We hope our rebuttal addressed your concerns. If not, please don't hesitate to reach out! We are willing to give more details.

Best,

Authors

---

### Decision · Program_Chairs · 2025-07-08

**Decision:**

Accept

**Comment:**

This paper introduces Extragradient Preference Optimization (EGPO), a novel algorithm for Nash Learning from Human Feedback that addresses non-transitive human preferences through game-theoretic optimization. EGPO achieves theoretical guarantees of last-iterate convergence while providing a practical implementation via an online IPO formulation that avoids nested optimization. Overall, all reviewers are positive about this paper. They appreciate its strong theoretical contributions, practical algorithmic innovations, and empirical validation. A few concerns are raised, including limited scalability, incomplete baseline comparisons, insufficient analysis of hyperparameter sensitivity and computational efficiency. However, it seems that most of these concerns have been adequately addressed by the authors in discussion.